# Covering K-Cliques in Billion-Scale Graphs

## Abstract

The k-clique structure in graphs has been investigated in various real-world applications, such as community detection in complex networks, functional module discovery in biological networks, and link spam detection in web graphs. Despite extensive research on $k$-clique enumeration, the large number of k-cliques in many graphs poses a challenge for practical application and computation. To address this, we explore the $k$-clique $\tau$-cover problem, a generalization of the vertex cover problem. The problem aims to find a small set of vertices that can effectively represent all k-cliques in the graph. We prove the NP-hardness of finding the minimum k-clique cover. We propose a hierarchical solution that computes a small cover without enumerating k-cliques. Extensive experiments on real-world graphs verify the efficiency and effectiveness of our solution.

## CCS Concepts

• **Mathematics of computing → Graph algorithms**.

## Keywords

clique, $k$-clique, clique cover, vertex cover, set cover

### ACM Reference Format:
Anonymous Author(s). 2018. Covering K-Cliques in Billion-Scale Graphs. In *Proceedings of Make sure to enter the correct conference title from your rights confirmation emai (Conference acronym 'XX)*. ACM, New York, NY, USA, 10 pages. https://doi.org/XXXXXXX.XXXXXXX

## 1 Introduction

Given a universe of elements and a collection of subsets whose union equals the universe, the set cover problem is to find the smallest sub-collection of these subsets whose union equals the universe. There have been extensive studies on variants of this problem in the context of the graph model, such as vertex cover [3–5, 11], $k$-path cover [2, 6, 13, 14] and maximal clique cover [25].

Given an undirected graph $G$, a $k$-clique is a subgraph of $G$ with $k$ vertices that are pairwise adjacent. The $k$-clique model has a wide range of real-world applications, such as identifying overlapping communities of complex networks in nature and society [28], discovering functionally related modules in gene (protein) association networks [1], link spam detection in web graphs [19, 29], and finding $k$-clique communities in mobile networks [15, 17]. While extensive research has been conducted on listing $k$-cliques, the number of $k$-cliques in many real-world graphs is often very large,

making a significant challenge in its applications. For instance, a graph might contain $O(3^{\frac{n}{3}})$ $k$-cliques [27], where $n$ is the number of vertices. Many $k$-cliques tend to overlap, and this overlap can be utilized to reduce the number of $k$-cliques in practical applications. Instead of directly using all $k$-cliques, an alternative approach leverages this overlap to select a subset of vertices that can represent all $k$-cliques.

We study the $k$-clique $\tau$-cover problem, which is a generalization of the vertex cover problem. Given a graph, a vertex set is called a $k$-clique $\tau$-cover of the graph if the vertex set overlaps every $k$-clique in the graph with at least $\tau$ vertices. Unlike the exponential number of $k$-cliques, the $k$-clique $\tau$-cover size is clearly bounded by the number of vertices. Additionally, the $k$-clique $\tau$-cover can ensure that a specific portion of every community is covered, which is useful in practical applications. For instance, in advertising, covering a subset of a clique can enable message propagation to eventually reach all its members, making this marketing strategy more cost-effective than reaching everyone at once. In particular, as shown in our case study Section 6.3, the $k$-clique $\tau$-cover outperforms the maximal clique cover [25] in group buying advertising [8].

In this paper, we investigate how to find a small $k$-clique $\tau$-cover. We prove that the optimization version of this problem - finding the minimum $\tau$-cover is NP-hard. To find a small cover, a straightforward approach is to precompute all $k$-cliques and then use a greedy algorithm to iteratively select high-degree vertices from each uncovered $k$-clique until all $k$-cliques are covered. To mitigate the expensive computational cost of precomputing all $k$-cliques, we first propose an improved approach that integrates $k$-clique listing and cover computation. It allows early termination of enumerating $k$-cliques based on certain heuristics. Despite pruning techniques, this method may be less effective when there are numerous cliques in large graphs, and there is no clear tight worst-case time complexity compared with the naive approach.

To further improve the efficiency, we observe that a vertex cover can be seen as a 2-clique cover, and we extend this observation to compute a $k$-clique $\tau$-cover. We propose a bottom-up hierarchical approach that computes the cover without enumerating $k$-cliques. Our method runs $k$ iterations to obtain a $k$-clique $\tau$-cover in $O(k \cdot m)$ time, where $m$ is the number of edges. This approach improves efficiency by avoiding the time-consuming $k$-clique enumeration process while maintaining correctness. In addition, we apply several pruning strategies to reduce the cover size. We also discuss ideas for updating the cover when the graph updates in the Appendix. We summarize our main contributions as follows.

- We formulate and study the $k$-clique $\tau$-cover problem. We prove the NP-hardness of finding the minimum $k$-clique $\tau$-cover. We present an improved approach to compute a small cover based on the $k$-clique listing.
- We propose an efficient hierarchical solution to compute a small cover without listing the $k$-cliques.
- We conduct extensive experiments on ten real-world graphs. The results verify the efficiency of our hierarchical solution.

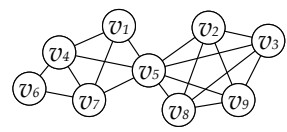

**Figure 1: An undirected graph $G$.**

## 2  Preliminaries

We study an undirected graph $G(V, E)$, where $V$ is the set of vertices and $E$ is the set of edges. We use $n$ and $m$ to denote $|V|$ and $|E|$, respectively. The set of neighbors of a given vertex $v$ is denoted by $N_v$. We assume that the graph has no self-loops and no duplicate edges. Given a graph $G(V, E)$, an induced subgraph of a vertex set $S$ (denoted by $G[S]$) includes all vertices in $S$ and all edges between them from $G$, i.e. $G[S] = (S, \{(u, v) \in E | u \in S \wedge v \in S\})$. A $k$-clique $C$ is a subgraph of $G$ with $k$ vertices that are pairwise adjacent. When the context is clear, we consider $C$ as a vertex set. The set of all $k$-cliques in $G$ is denoted by $C(G, k)$.

*Definition 2.1 (k-Clique $\tau$-Cover and k-Clique Cover).* Given a graph $G$, a vertex set $S$ is called a $k$-clique $\tau$-cover of $G$ if $\forall C \in C(G, k)$, we have $|C \cap S| \geq \tau$. We call $S$ a $k$-clique cover if $\tau = 1$.

In this paper, given a graph $G$, two integers $k$ and $\tau$, we aim to compute a small vertex set $S$ that is a $k$-clique $\tau$-cover of $G$.

*Example 2.2.* Figure 1 shows an undirected graph $G$. Given $k = 4$, the 4-cliques in $G$ are $\{v_1, v_4, v_5, v_7\}$, $\{v_2, v_3, v_5, v_8\}$, $\{v_2, v_3, v_5, v_9\}$, $\{v_2, v_3, v_8, v_9\}$, $\{v_2, v_5, v_8, v_9\}$, and $\{v_3, v_5, v_8, v_9\}$. The vertex set $\{v_2, v_5\}$ forms a 4-clique cover of $G$. If $\tau = 2$, the vertex set $\{v_2, v_3, v_5, v_7\}$ forms a 4-clique 2-cover of $G$.

THEOREM 2.3. *Computing the minimum $k$-clique $\tau$-cover for any $k \geq 2$ and $\tau \geq 1$ is NP-hard.*

PROOF. For $\tau = 1$, the problem is equivalent to the well-known NP-hard hitting set problem [20]. Given a collection of subsets of $V$, the hitting set problem is to find the smallest subset $S \in V$ that intersects every set in the collection. In this context, each $k$-clique corresponds to a subset of $V$. For $\tau > 1$, the problem remains NP-hard, as it can be reduced from the set cover problem, which is also NP-hard [20, 25]. □

Note that finding the minimum $\tau$-cover for maximal cliques is also NP-hard [25].

## 3  Related Work

### 3.1  K-Clique Enumeration

The problem of listing $k$-cliques has been extensively studied. Chiba and Nishizeki [9] introduced the first practical algorithm for listing $k$-cliques. Building on their work, Finocchi et al. [12] improved the algorithm using a degree ordering technique. Danisch et al. [10] refined the algorithm with a degeneracy ordering technique. Li et al. [24] later developed a hybrid algorithm that combines degeneracy and color ordering techniques. Additionally, Yuan et al. [35] increased the algorithm efficiency by implementing single instruction multiple data (SIMD) instructions. Most recently, Wang

et al. [33] proposed a branch-and-bound algorithm incorporating an edge-oriented branching strategy.

In addition to the $k$-clique listing problem, the $k$-clique densest subgraph problem has received much attention [16, 31, 32, 36]. This problem aims to find the subgraph with the highest $k$-clique density, defined as the ratio of the number of $k$-cliques to the number of vertices in it [32]. Sun et al. [31] developed a simple algorithm to find $k$-clique densest subgraphs. However, the algorithm is not scalable for large $k$ values and large-scale graphs because it needs to list all $k$-cliques repeatedly in each iteration. To alleviate this issue, He et al. [16] proposed an index structure that accelerates $k$-clique listing based on the succinct clique tree [18]. Additionally, Zhou et al. [36] proposed a framework that relies on $k$-clique counting rather than listing, which is usually much faster.

### 3.2  Other Covering Problems

Several other covering problems have been explored in the literature. Karp [20] formulated the vertex cover problem and established it as one of the fundamental NP-complete problems. Since then, many heuristics have been proposed to address this problem [3, 5, 11]. Notably, Angel et al. [4] provided a comprehensive evaluation of various heuristics for the vertex cover problem. Bresar et al. [6] introduced the problem of finding the minimum $k$-path cover. Subsequently, Funke et al. [13, 14] devised efficient algorithms to compute small $k$-path covers and explored various application scenarios. Furthermore, Akiba et al. [2] proposed a hierarchical approach to compute and dynamically maintain $k$-path covers. Recently, Li et al. [25] studied the problem of finding a vertex set that approximately covers all maximal cliques with a given coverage threshold.

## 4  Covering By Listing K-Cliques

A straightforward solution is to precompute all $k$-cliques. Then, for each uncovered $k$-clique, we iteratively select the highest-degree vertices within the clique into the cover set until the clique is covered. However, listing all $k$-cliques is time-consuming. In this section, we propose an algorithm to identify the cover by avoiding listing all $k$-cliques. Section 4.1 reviews the algorithm for $k$-clique listing. Section 4.2 proposes the algorithm for computing $k$-clique $\tau$-cover.

### 4.1  The KClist Algorithm

The state-of-the-art algorithm for $k$-clique listing is called KClist [10]. The algorithm is based on a vertex order and adopts a depth-first search paradigm. The pseudocode is presented in Algorithm 1. Given a vertex order, the algorithm first generates a directed acyclic graph (DAG) $\overrightarrow{G}$ by linking each vertex to its lower-ranking neighbors (line 1). Several orders have been investigated to improve the efficiency of $k$-clique listing including degree, degeneracy, or other graph metrics [24]. The KClist algorithm uses the degeneracy order to minimize the number of out-neighbors for each vertex in the DAG, which helps reduce the search space in the algorithm. We will discuss the effect of different ordering strategies on our problem in the next section. The algorithm invokes a recursive procedure, ProcKCL (line 2) to enumerate $k$-cliques based on the DAG.

**Algorithm 1:** KClist($G, k$)

   **Input:** A graph $G(V, E)$ and a positive integer $k$

   **Output:** All k-cliques in $G$

1  $\overrightarrow{G} \leftarrow$ a DAG generated by a total ordering on $V$;

2  ProcKCL($\overrightarrow{G}, \emptyset, k$);

3  **Procedure** ProcKCL($\overrightarrow{G}, C, k$)

4    **if** $k = 2$ **then**

5       **foreach** edge $\langle u, v \rangle \in E(\overrightarrow{G})$ **do**

6          output a k-clique $C \cup \{u, v\}$;

7    **else**

8       **foreach** vertex $v \in V(\overrightarrow{G})$ **do**

9          $\overrightarrow{G}_v \leftarrow \overrightarrow{G}[N_v^+]$;

10         ProcKCL($\overrightarrow{G}_v, C \cup \{v\}, k - 1$);

ProcKCL takes the following input parameters: a DAG $\overrightarrow{G}$, a partial clique $C$ and a positive integer $k$. When $k \neq 2$ in ProcKCL (lines 7–10), it processes each vertex by recursively creating a subgraph $\overrightarrow{G}_v$ induced by its outgoing neighbors $N_v^+$ and adding the vertex to the current partial clique. When $k = 2$ in ProcKCL (lines 4–6), it iterates over all edges in the subgraph to form and output $k$-cliques. The worst-case time complexity of the KClist algorithm is $O(km(\delta/2)^{k-2})$, where $\delta$ is the graph degeneracy.

## 4.2 Our Approach: KCCB

We propose an algorithm that integrates the process of computing covering vertices with the KClist algorithm. Our approach utilizes the intermediate cover set during the computation and certain pruning techniques to enable early termination of enumerating $k$-cliques. By doing so, we effectively reduce the search space of unexplored cliques that have already been covered. The pruning technique is based on the cover lower bound defined as follows.

*Definition 4.1 (Cover Lower Bound).* Given a partial cover set $\mathcal{S}$ and a vertex set $C$, the cover lower bound of $C$ is the number of vertices in $C$ that are also in $\mathcal{S}$, i.e., $|C \cap \mathcal{S}|$.

Based on Definition 4.1, when we find a covered partial clique (i.e., its cover lower bound is not less than $\tau$), the algorithm can safely stop expansion from the partial clique since all $k$-cliques containing the partial clique must have been covered. When the algorithm finds an uncovered $k$-clique (i.e., its cover lower bound is less than $\tau$), the algorithm picks certain vertices to cover the $k$-clique.

We present our algorithm called KCCB with the lower-bound-based pruning technique in Algorithm 2. Compared to KClist, KCCB achieves higher practical efficiency given the pruning techniques. We first generate a DAG using the degree ordering on $V$ (line 1). We choose the degree ordering because we wish to start with low-degree vertices and postpone the more complex computations associated with high-degree vertices. During the computation, we prioritize the high-degree vertices as the cover, thus reducing a significant portion of the high-degree related computation. When $k = 0$ (lines 4–8), after listing an uncovered $k$-clique, we iteratively

**Algorithm 2:** KCCB($G, k, \tau$)

   **Input:** A graph $G(V, E)$, two positive integers $k$ and $\tau$

   **Output:** A $k$-clique $\tau$-cover $\mathcal{S}$

1  $\overrightarrow{G} \leftarrow$ a DAG generated by the degree ordering on $V$;

2  ProcKCC($\overrightarrow{G}, \emptyset, k, \emptyset$);

3  **Procedure** ProcKCC($\overrightarrow{G}, C, k, \mathcal{S}$)

4    **if** $k = 0$ **then**

5       **foreach** vertex $v \in C$ in decreasing degree order **do**

6          $C \cup \{v\}, \mathcal{S} \leftarrow \mathcal{S} \cup \{v\}$;

7          Compute $\underline{\tau}$ by Equation (1);

8          **if** $\underline{\tau} \geq \tau$ **then** break;

9    **else**

10       **foreach** vertex $v \in V(\overrightarrow{G})$ **do**

11          $C \cup \{v\}, \overrightarrow{G}_v \leftarrow \overrightarrow{G}[N_v^+]$;

12          Compute $\underline{\tau}$ by Equation (1), $C \setminus \{v\}$;

13          **if** $\underline{\tau} \geq \tau$ **then** continue;

14          ProcKCL($\overrightarrow{G}_v, C \cup \{v\}, k - 1, \mathcal{S}$);

add vertices from the $k$-clique in decreasing degree order until the cover lower bound satisfies the predefined threshold $\tau$. We prioritize vertices with higher degrees here to reduce the cover size as they are more likely to appear in multiple $k$-cliques. When $k \neq 0$ (lines 9–14), after adding a vertex to the current partial clique, we compute the cover lower bound to determine whether to continue the computation (lines 12–13). Based on the observation on the KClist enumeration process, we derive the following lemma and equation for the cover lower bound.

LEMMA 4.2. *Given a cover set $\mathcal{S}$, a partial clique $C$ and a positive integer $k$, for any $k$-clique extended by $C$, its cover lower bounds $\underline{\tau}$ satisfy:*

$$\underline{\tau} \geq |C \cap \mathcal{S}| + \max\{0, k - |C| - |N_v^+ - \mathcal{S}|\} \quad (1)$$

In this equation, $C$ represents the current partial clique, $\mathcal{S}$ represents the current cover set, and $N_v^+$ denotes the outgoing neighbors of the last vertex $v$ added to $C$. The cover lower bound consists of two parts. The first one $|C \cap \mathcal{S}|$ computes the current coverage. The second one $\max\{0, k - |C| - |N_v^+ - \mathcal{S}|\}$ computes the minimum possible coverage gain in the unexplored subgraph. To form a $k$-clique, we need $k - |C|$ additional vertices from the vertex set $N_v^+$ of the unexplored subgraph. $|N_v^+ - \mathcal{S}|$ represents the number of vertices in the unexplored subgraph that are not in the current cover set. To compute the minimum possible coverage gain, we first determine the maximum possible number of vertices in the $k$-clique that are not in the current cover set in the worst case. This is given by $|N_v^+ - \mathcal{S}|$. The remaining vertices needed to form the $k$-clique must be in the current cover set, and the number of these vertices represents the minimum possible coverage gain in the unexplored subgraph. Lastly, we subtract the number of vertices in the unexplored subgraph but not in the current cover set from $k - |C|$ and use the max function to ensure the final value is non-negative.

*Example 4.3.* Figure 2 shows a running example of Algorithm 2 for $k = 4$ and $\tau = 2$. Shaded vertices represent those included

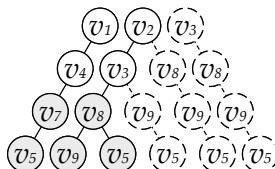

Figure 2: A running example of KCCB.

in the cover, while vertices surrounded by dashed lines indicate those excluded by our pruning strategy. For instance, given $S = \{v_5, v_7, v_8, v_9\}$, $C = \{v_2, v_3, v_9\}$, $v = v_9$, and $N_v^+ = \{v_5\}$, we have $\underline{\tau} \geq 2 = \tau$ according to Equation (1), and we can stop further expansion from $C$ to $\{v_2, v_3, v_9, v_5\}$. As shown in the figure, this pruning strategy effectively reduces the enumeration of $k$-cliques by more than half.

The main limitation of Algorithm 2 lies in the computational bottleneck of listing $k$-cliques. Although the method incorporates a pruning strategy, its effectiveness can vary considerably in different datasets and mainly depends on the degree of overlap between $k$-cliques and the number of $k$-cliques in the graph. When there is a small overlap between $k$-cliques or numerous $k$-cliques, the computation will be extremely slow because it needs to list all $k$-cliques in the worst case. This limitation inherently restricts the scalability of the method, particularly for large and complex graphs.

## 5 The Hierarchical Approach

To improve the efficiency of covering $k$-cliques, we aim to develop an algorithm without listing any $k$-cliques. To this end, we first focus on the $k$-clique covering problem (i.e., $\tau = 1$) and propose a hierarchical approach in Section 5.1. We then present the algorithm in Section 5.2. After that, we extend our solution to the $k$-clique $\tau$-cover problem (i.e., $\tau \geq 1$).

### 5.1 $k$-Clique Cover Hierarchy

Given an undirected graph $G(V, E)$, a vertex cover is a subset $R \subseteq V$ such that for every edge $\langle u, v \rangle \in E$, we have $u \in R$ or $v \in R$. Extensive studies have been conducted to compute a small vertex cover [3, 5, 11]. The vertex cover is clearly a 2-clique cover since each edge is a 2-clique. Our intuition is to start from the 2-clique cover and iteratively extend a $k$-clique cover to a $(k + 1)$-clique cover.

*Example 5.1.* Figure 3 shows an example of a vertex cover and its induced subgraph. The shaded vertices in Figure 3(a) are the vertex cover of $G$. Figure 3(b) is the subgraph induced by the vertex cover.

Our idea is that the vertex cover of the subgraph induced by a $k$-clique cover forms a $(k + 1)$-clique cover of the original graph. The following lemma and theorem formalize this idea.

Lemma 5.2. *Given a $(k+1)$-clique, at least two vertices are required to cover all $k$-cliques within the $(k + 1)$-clique.*

Proof. A $(k + 1)$-clique contains $k$ distinct $k$-cliques, each differing by exactly one vertex. If only one vertex is used to cover the $k$-clique, there will be a $k$-clique that is not covered due to

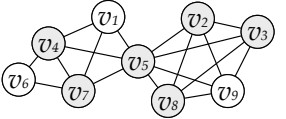
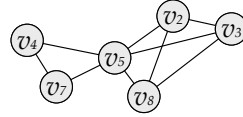

(a) A vertex cover of $G$.  (b) The subgraph induced by the vertex cover of $G$.

Figure 3: Example of a vertex cover and its induced subgraph.

this different vertex. Therefore, at least two vertices are needed to ensure that all $k$-cliques are covered. □

Theorem 5.3. *Given a graph $G$ and its $k$-clique cover $S$, the vertex cover of the induced subgraph $G[S]$ is a $(k + 1)$-clique cover of $G$.*

Proof. By Lemma 5.2, at least two vertices are needed to cover the $k$-cliques within a $(k + 1)$-clique. Every two vertices share an edge because they belong to the same $(k + 1)$-clique. Based on this observation, we compute the vertex cover of $G[S]$, ensuring that at least one endpoint of each edge is included. This guarantees that the resulting cover forms a $(k + 1)$-clique cover, which has a smaller size compared to the $k$-clique cover. □

Motivated by Theorem 5.3, we first compute a vertex cover of the original graph, which forms a 2-clique cover since each edge represents a 2-clique. We then iteratively compute the corresponding induced subgraph and vertex cover to obtain a $k$-clique cover for each subsequent $k$. Based on this idea, we define a $k$-clique cover hierarchy, which represents the iterative process of constructing a $k$-clique cover, consisting of a sequence of graphs and vertex sets. Formally, we define the $k$-clique cover hierarchy as follows.

*Definition 5.4 (k-Clique Cover Hierarchy).* A $k$-clique cover hierarchy $\mathcal{H}(\mathcal{G}, \mathcal{R})$ of a graph $G(V, E)$ consists of a sequence of graphs $\mathcal{G} = (G_0, G_1, G_2, ..., G_{k-1})$ and a sequence of vertex sets $\mathcal{R} = (R_0, R_1, R_2, ..., R_{k-1})$ that satisfy the following conditions.

(1) $G_0 = G, R_0 = V$.
(2) $R_i$ is a vertex cover of $G_{i-1}$ for $1 \leq i \leq k - 1$.
(3) $G_i = G[R_i]$ for $1 \leq i \leq k - 1$.

The following theorem generalizes Theorem 5.3 in the context of the $k$-clique cover hierarchy.

Theorem 5.5. *Given a graph $G(V, E)$ and its $k$-clique cover hierarchy $\mathcal{H}(\mathcal{G}, \mathcal{R})$. For any induced subgraph $G_i = G[R_i]$, the vertex cover $R_i$ of $G_{i-1}$ is an $(i + 1)$-clique cover of $G$, where $1 \leq i \leq k - 1$.*

Proof. We prove this theorem by induction on $i$. For $i = 1$, $G_0 = G$ and $R_0 = V$. The vertex cover $R_1$ of $G_0$ covers all edges in $G$, implying $R_1$ forms a 2-clique cover since each edge is a 2-clique. Assume that for certain $i \geq 1$, $R_i$ is an $(i + 1)$-clique cover of $G$. By Theorem 5.3, $R_{i+1}$, which is the vertex cover of the induced subgraph $G_i = G[R_i]$, forms an $(i + 2)$-clique cover of $G$. By induction, for all $1 \leq i \leq k - 1$, the vertex cover $R_i$ of $G_{i-1}$ is an $(i + 1)$-clique cover of $G$. □

*Example 5.6.* Figure 4 shows a $k$-clique cover hierarchy where $k = 3$. The hierarchy consists of three graph layers, $G_0$, $G_1$, and $G_2$, from bottom to top. In each layer, the shaded vertices represent

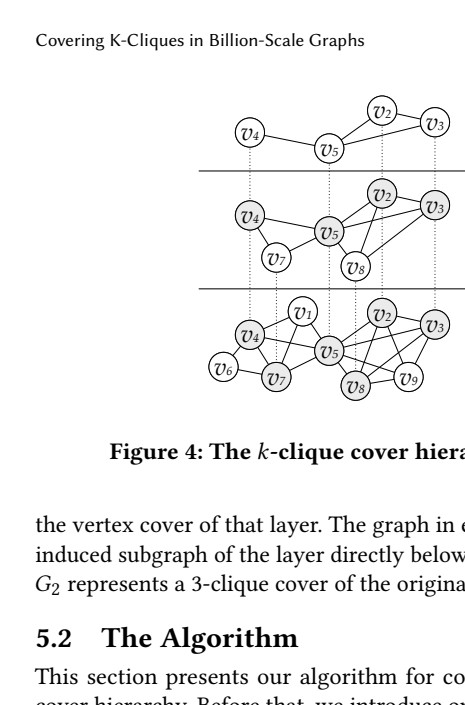

**Figure 4: The $k$-clique cover hierarchy $\mathcal{G}$ of $G$.**

the vertex cover of that layer. The graph in each upper layer is an induced subgraph of the layer directly below it. The topmost layer $G_2$ represents a 3-clique cover of the original graph $G$.

## 5.2 The Algorithm

This section presents our algorithm for constructing a $k$-clique cover hierarchy. Before that, we introduce optimizations to reduce the cover size while maintaining correctness. We apply two existing techniques (core number [7, 23, 30] and graph color [24, 34]) to the $k$-clique cover. We observe that certain vertices in the cover set do not belong to any $k$-cliques or are part of $k$-cliques already covered by other vertices. Core number pruning identifies vertices unlikely to belong to any $k$-cliques. The core number of a vertex is the largest integer $k$ such that the vertex is part of a $k$-core, a maximal subgraph where every vertex has at least degree $k$ [30]. Since every $k$-clique forms a $(k-1)$-core, we reduce the candidate set by pruning vertices whose core number is less than $k-1$. This computation can be completed in $O(m)$ time [7, 23]. Graph color pruning further refines the $k$-clique cover. In this pruning strategy, adjacent vertices are assigned different colors, and because every $k$-clique must have at least $k$ distinct colors, we prune vertices with insufficient colors in their neighborhoods. This technique can be performed in $O(m)$ time using a greedy algorithm [34]. Specifically, for each vertex $v \in \mathcal{S}$, we count the number of distinct colors in its neighborhood $N_v$ that are not part of the $k$-clique cover $\mathcal{S}$. This count is represented as $\chi(N_v \setminus \mathcal{S})$.

LEMMA 5.7. *Given a $k$-clique cover $\mathcal{S}$ and a coloring of $G[V \setminus \mathcal{S}]$, let $v$ be a vertex in $\mathcal{S}$ such that $\chi(N_v \setminus \mathcal{S}) < k - 1$, then $\mathcal{S} \setminus v$ remains a valid $k$-clique cover.*

PROOF. We consider two cases for the vertex $v$: (1) $v$ is the only vertex chosen in a $k$-clique. The removal of $v$ from $\mathcal{S}$ is not possible because $\chi(N_v \setminus \mathcal{S}) \geq k - 1$. (2) $v$ is not the only vertex chosen in a $k$-clique. When $v$ is removed from $\mathcal{S}$, we assign a minimal color to $v$ based on $N_u \setminus \mathcal{S}$. Consequently, $\chi(N_u \setminus \mathcal{S})$ may increase by one for any $u \in \mathcal{S} \cap N_v$. This reassignment of colors ensures that at least one vertex remains to cover each $k$-clique, thereby maintaining the validity of $\mathcal{S} \setminus v$ as a $k$-clique cover. □

The pseudocode for graph color pruning is presented in Algorithm 3. We first assign minimal colors to the vertices in $V \setminus \mathcal{S}$ based on the induced subgraph $G[V \setminus \mathcal{S}]$ (line 1). For each vertex $v$ in the $k$-clique cover $\mathcal{S}$, we check the number of different colors

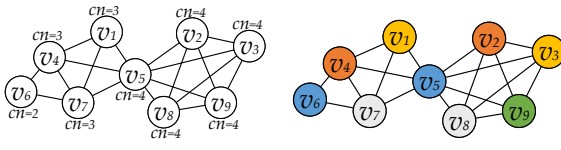

**(a) Core number pruning**  **(b) Graph color pruning**

**Figure 5: Illustration of two pruning strategies.**

---

**Algorithm 3:** ColorPruning$(G, k, \mathcal{S})$

**Input:** A graph $G(V, E)$, a positive integer $k$ and a $k$-clique cover $\mathcal{S}$

**Output:** A pruned $k$-clique cover

1 assign minimal colors to $V \setminus \mathcal{S}$ based on $G[V \setminus \mathcal{S}]$;

2 **foreach** $v \in \mathcal{S}$ **do**

3   **if** $\chi(nb_v \setminus \mathcal{S}) < k - 1$ **then**

4     $\mathcal{S} \leftarrow \mathcal{S} \setminus v$;

5     assign a minimal color to $v$ based on $N_v \setminus \mathcal{S}$;

6 **return** $\mathcal{S}$

---

in its neighborhood, excluding any neighbors that are part of the $k$-clique cover (line 3). If the number of different colors is greater than or equal to $k - 1$, we retain the vertex in the $k$-clique cover. Otherwise, we remove the vertex from the $k$-clique cover and assign it a minimal color (lines 4-5). We then proceed to the next vertex (line 2).

*Example 5.8.* Figure 5(a) shows the core number of each vertex. Figure 5(b) shows the minimal graph coloring.

Our final algorithm for constructing a $k$-clique cover hierarchy is detailed in Algorithm 4. Based on the core number pruning strategy, we initialize the first layer with the original graph and its vertices whose core number is greater than or equal to $k - 1$ (line 2). We then iteratively compute the vertex cover of the previous layer's graph and use it to construct the induced subgraph for the current layer (lines 4-5). It is worth noting that when constructing the new induced subgraph, we base it on the preceding induced subgraph rather than the original graph. This approach is more efficient because the new induced subgraph is a subgraph of the preceding one, which reduces the graph size and improves construction efficiency (line 5). Based on Theorem 5.3 and Theorem 5.5, we repeat this process for $k - 1$ iterations to obtain the $k$-clique cover of the original graph (line 4). After obtaining the $k$-clique cover, we call Algorithm 3 to refine it (line 6).

LEMMA 5.9. *The time complexity of Algorithm 4 is $O(k \cdot m)$.*

PROOF. The core number computation and pruning take $O(m)$ time. The $k - 1$ iterations of vertex cover computation and induced subgraph construction take $O(k \cdot m)$ time. Finally, the minimal graph coloring and pruning take $O(m)$. Therefore, the total time complexity is $O(k \cdot m)$. □

A key step in our hierarchical approach is finding a high-quality (small) vertex cover for a graph $G(V, E)$. Although the minimum

---

**Algorithm 4:** KCC($G, k, \tau$)

**Input:** A graph $G(V, E)$ and a positive integer $k$
**Output:** A $k$-clique cover of $G$
1  compute the $(k - 1)$-core of $G$;
2  $G_0 \leftarrow G, R_0 \leftarrow \{v | v \in V : core(v) \geq k - 1\}$;
3  **for** $i \leftarrow 1, 2, ..., k - 1$ **do**
4       $R_i \leftarrow$ VertexCover($G_{i-1}$);
5       $G_i \leftarrow$ the induced subgraph of $R_i$ in $G_{i-1}$;
6  **return** ColorPruning($G[R_0], k, R_{k-1}$)

---

vertex cover problem is NP-complete [20], several heuristics have been proposed to find a small vertex cover [3–5, 11]. Let $R$ be the vertex cover under construction. For each $u \in V$, we consider the following heuristics:
- **LR:** Add $N_v$ to $R$ if $v \notin R$.
- **LL:** Add $u$ to $R$ if $\{v | v \in N_u \wedge v \notin R \wedge v > u\} \neq \emptyset$.
- **SLL:** Add $u$ to $R$ if $\{v | v \in N_u \wedge v \notin R \wedge [deg(v) < deg(u) \vee (deg(v) = deg(u) \wedge v > u)]\} \neq \emptyset$.

Among these heuristics, LR does not use any vertex ordering, LL uses simple vertex ID ordering, and SLL uses decreasing degree ordering. In our approach, we employ the reverse degeneracy (RD), as described in [2], which prioritizes vertices with the most uncovered edges. We compare these heuristics in our experiments.

## 5.3  Extending to $k$-Clique $\tau$-Cover

In this section, we extend our solution for $k$-clique cover to the $k$-clique $\tau$-cover problem (i.e., $\tau \geq 1$). We examine the relationship between a $\tau$-cover ($\tau = 1$) and a $\tau$-cover ($\tau \geq 1$) and observe that a $k$-clique consists of $k$ $(k - 1)$-cliques. The intuition is to compute a $k$-clique $\tau$-cover based on the $(k-1)$-clique cover. Specifically, if we have a cover for all $(k - 1)$-cliques within all $k$-cliques, this cover will also be a $k$-clique 2-cover. For example, a 2-clique cover is also a 3-clique 2-cover because a 3-clique consists of three 2-cliques, and we need at least two vertices to cover these 2-cliques within the 3-clique. The following lemma formalizes and generalizes this observation.

LEMMA 5.10. *Given a graph $G$, a $k$-clique $\tau$-cover of $G$ is also a $(k + 1)$-clique $(\tau + 1)$-cover of $G$.*

PROOF. A $(k+1)$-clique contains $k$ $k$-cliques, each differing from the others by exactly one vertex. Therefore, the $\tau$-cover of all these $k$-cliques must contain at least $\tau + 1$ vertices. This is because there is a pair of cliques that differ by one vertex, one of which is in the cover set and the other is not. Consequently, an additional vertex is required to cover all $k$-cliques, resulting in a minimum of $\tau + 1$ vertices in the cover set, which forms a $(k + 1)$-clique $(\tau + 1)$-cover. □

Based on Lemma 5.10, we can derive a $k$-clique $\tau$-cover of a graph by computing the $(k - \tau + 1)$-clique cover. The pseudocode is presented in Algorithm 5. We initialize the first layer with the original graph and its vertices whose core number is greater than or equal to $k - \tau$ (line 2). Next, we iteratively compute the vertex cover for each intermediate graph and construct the corresponding induced subgraph (lines 4-5). This process is repeated for $k - \tau$

---

**Algorithm 5:** KTCC($G, k, \tau$) - Ours

**Input:** A graph $G$, two positive integers $k$ and $\tau$
**Output:** A $k$-clique $\tau$-cover of $G$
1  compute the $(k - \tau)$-core of $G$;
2  $G_0 \leftarrow G, R_0 \leftarrow \{v | v \in V : core(v) \geq k - \tau\}$;
3  **for** $i \leftarrow 1, 2, ..., k - \tau$ **do**
4       $R_i \leftarrow$ VertexCover($G_{i-1}$);
5       $G_i \leftarrow$ the induced subgraph of $R_i$ in $G_{i-1}$;
6  **return** $R_{k-\tau}$

---

Table 1: Statistics of datasets.

| Name | $n$ | $m$ | Type |
|---|---|---|---|
| *Email* | 1,005 | 25,571 | Communication |
| *EgoFacebook* | 4,039 | 88,234 | Social |
| *MusaeFacebook* | 22,470 | 171,002 | Social |
| *Epinions* | 75,879 | 508,837 | Social |
| *EgoTwitter* | 81,306 | 1,768,149 | Social |
| *BerkStan* | 685,230 | 7,600,595 | Web |
| *LiveJournal* | 4,847,571 | 68,993,773 | Social |
| *Orkut* | 3,072,441 | 117,185,083 | Social |
| *IT-2004* | 41,291,594 | 1,150,725,436 | Web |
| *Friendster* | 65,608,366 | 1,806,067,135 | Social |

iterations (line 3). Finally, we return the $(k - \tau + 1)$-clique cover, which is also a $k$-clique $\tau$-cover. The time complexity of Algorithm 5 is $O((k - \tau) \cdot m)$.

## 6  Experimental Evaluation

All algorithms are implemented in C++ and compiled with the g++ compiler at the -O3 optimization level. All experiments are conducted on a Linux machine with dual Intel Xeon Gold 6342 2.8GHz CPUs and 512GB RAM. We evaluate the algorithms on ten real-world graphs. Detailed dataset statistics are given in Table 1, where $n$ is the number of vertices and $m$ is the number of edges. All datasets are from SNAP[1] and NR[2]. In the following experiments, we omit the results of the naive method which precomputes all $k$-cliques, as it is consistently at least an order of magnitude slower than KCCB, despite producing similar cover sizes. Additionally, the naive method fails to process more than half of the datasets due to memory limitations or exceeding the 12-hour time limit.

### 6.1  $k$-Clique Cover

*6.1.1  Running Time.* In this experiment, we evaluate the efficiency of different algorithms for the $k$-clique cover (i.e., $\tau = 1$), including KCCB and KCC. Regarding the input parameter, we vary the integer $k$ as 4, 5, 6, 7, 8, and 9 for each dataset. Figure 6 reports the running time of the algorithms. We can see that KCC is on average over one order of magnitude faster than KCCB. For example, in the EgoFacebook dataset, KCC has an average runtime of about 0.02 seconds, whereas KCCB takes around 1.35 seconds. Similarly, in

---

[1]https://snap.stanford.edu/
[2]https://networkrepository.com/

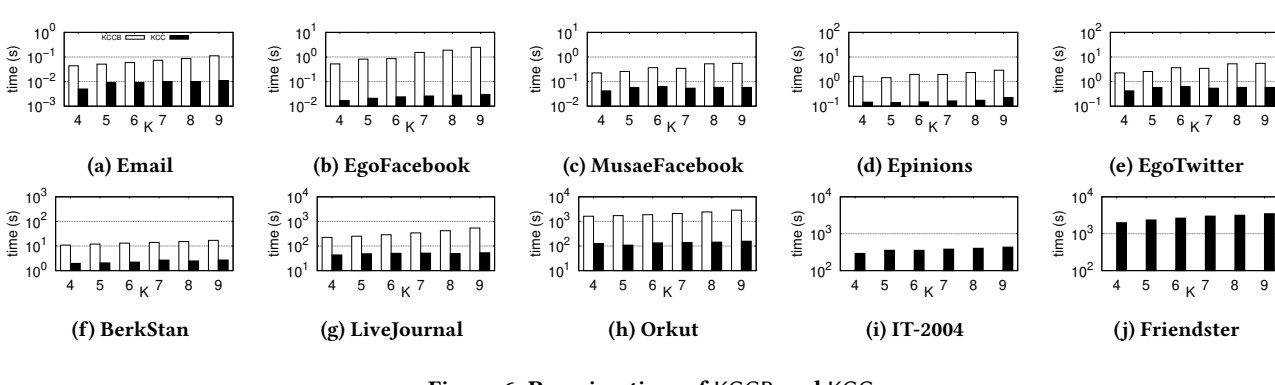

**Figure 6: Running time of KCCB and KCC.**

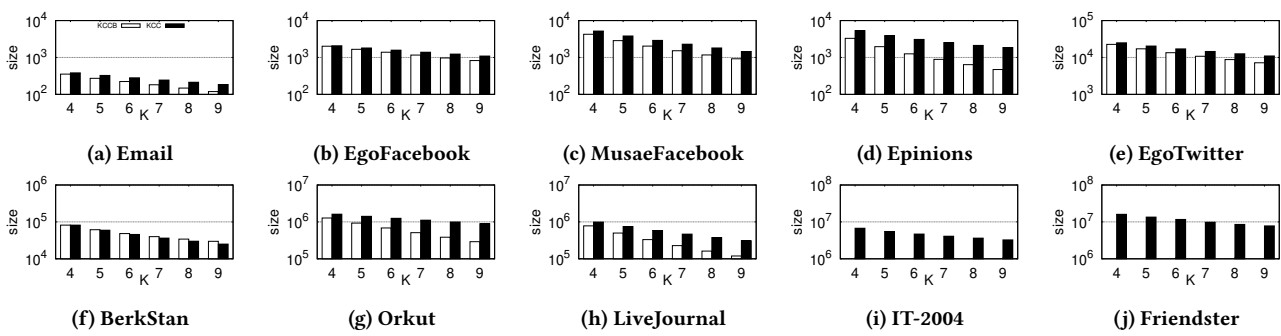

**Figure 7: Cover size of KCCB and KCC.**

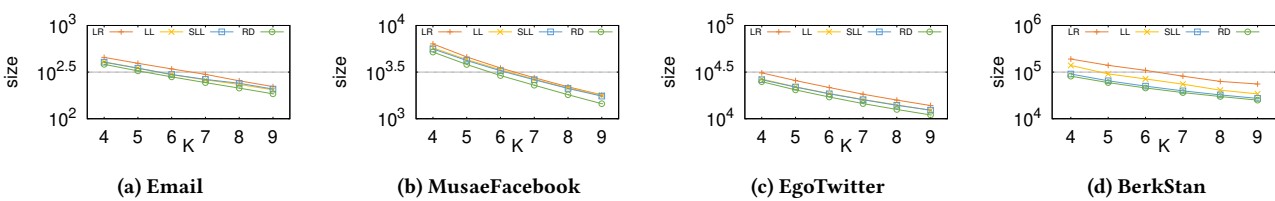

**Figure 8: Cover size of KCC under different vertex cover heuristics.**

the EgoTwitter dataset, KCC has an average runtime of about 2.47 seconds, while KCCB takes around 107.32 seconds. We notice that as $k$ increases, the runtime of KCC and KCCB increases. This is because as $k$ increases, KCC needs to compute more layers, and KCCB needs to process more $k$-cliques. Note that KCCB fails to process two billion-scale datasets in 12 hours.

*6.1.2 Cover Size.* In this experiment, we evaluate the $k$-clique cover size of different algorithms. Figure 7 reports the cover size of the algorithms. We can see that KCC and KCCB have similar cover sizes. For example, in the BerkStan dataset, when $k = 6$, the cover size of KCC is 45,689, while the cover size of KCCB is 48,253. We notice that as $k$ increases, the cover size of KCC and KCCB decreases because the degree of overlap between the $k$-cliques increases. We also evaluate the $k$-clique cover size of KCC under different vertex cover heuristics. Figure 8 reports the cover size of KCC with different vertex cover heuristics on four representative datasets. We can see that the cover size of KCC under RD is the smallest in all cases.

## 6.2 $k$-Clique $\tau$-Cover

*6.2.1 Running Time.* In this experiment, we evaluate the efficiency of different algorithms for the $k$-clique $\tau$-cover (i.e., $\tau \geq 1$), including KCCB and KTCC. Regarding the input parameters, we fix the integer $k$ to 9 and vary the predefined threshold $\tau$ as 1, 2, 3, 4, and 5 for each dataset. Figure 9 reports the running time of the algorithms on four representative datasets. We can see that KTCC is on average over one order of magnitude faster than KCCB. For example, in the EgoTwitter dataset, KTCC has an average runtime of about 0.14 seconds, whereas KCCB takes around 71.20 seconds. Similarly, in the BerkStan dataset, KTCC has an average runtime of about 0.63 seconds, while KCCB takes around 67.06 seconds. We notice that as $\tau$ increases, the runtime of KCCB increases because it needs to process more vertices to cover the $k$-cliques. We also notice that as $\tau$ increases, the runtime of KTCC decreases because it needs to compute fewer layers.

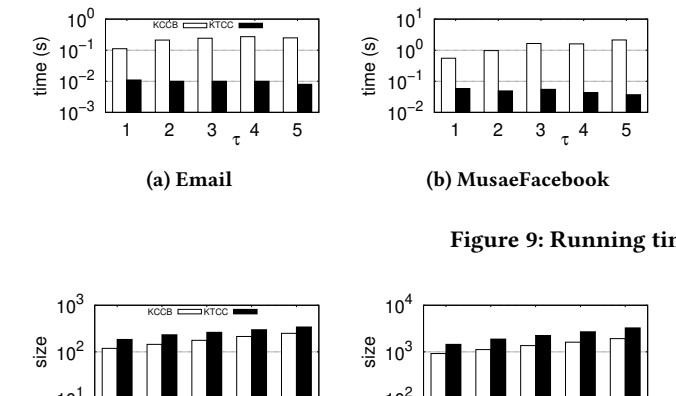
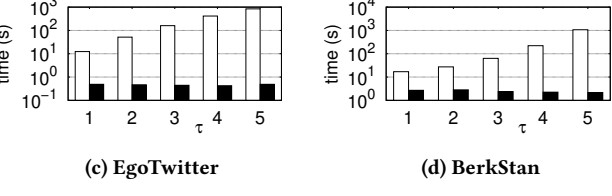

(a) Email  (b) MusaeFacebook  (c) EgoTwitter  (d) BerkStan

Figure 9: Running time of KCCB and KTCC.

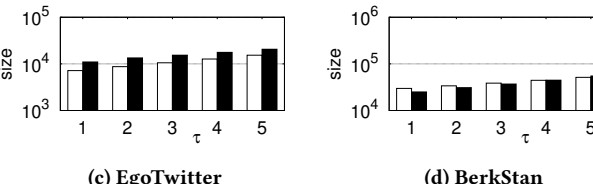

(a) Email  (b) MusaeFacebook  (c) EgoTwitter  (d) BerkStan

Figure 10: Cover size of KCCB and KTCC.

**Table 2: Vertex percentage and runtime for group buying advertising via clique covers on YouTube network.**

| Metric | MCC | EMCC | KCC ($k=3$) | KCC ($k=4$) | KCC ($k=5$) | KCC ($k=6$) |
|---|---|---|---|---|---|---|
| Vertices (%) | 25.0% | 4.5% | 6.9% | 2.9% | 1.6% | 1.0% |
| Runtime (s) | 181.3 | 128.8 | 2.2 | 2.6 | 2.9 | 3.4 |

6.2.2 *Cover Size.* In this experiment, we evaluate the $k$-clique $\tau$-cover size of different algorithms. Figure 10 reports the cover size of the algorithms on four representative datasets. We can see that KTCC and KCCB have similar cover sizes. For example, in the BerkStan dataset, when $\tau = 3$, the cover size of KTCC is 36,942, while the cover size of KCCB is 38,623. We notice that as $\tau$ increases, the cover size of KTCC and KCCB increases because more vertices are required to achieve a higher coverage of each $k$-clique.

## 6.3 Case Study

Online group buying is a business model that enables consumers to obtain products at discounted prices by purchasing in bulk [8]. In this model, a collective of individuals engages with sellers to secure these discounts. Sellers aiming to promote group-buying deals seek to target specific cliques of interconnected potential buyers above a certain size, as these groups are more likely to share similar purchasing decisions. Given that the cost of advertising increases with the number of targeted users (e.g., offering incentives such as free items or significant discounts), strategic targeting becomes crucial [21, 22, 26, 37]. In addition, research on group buying highlights the importance of member influence in advertising adoption [8]. Specifically, the opinions of group members regarding a product can significantly affect the purchasing behavior of others in the clique, with the influence increasing as more group members are exposed to the advertisement. Therefore, advertisers must ensure that a sufficient proportion of each clique is effectively targeted while minimizing the overall number of individuals approached.

The recent work has used maximal clique $\tau$-cover to address the group buying advertising problem with noticeable results [25]. In our work, we apply our hierarchical approach KCC to solve the same problem and demonstrate improved performance. We conduct our evaluation on the YouTube social network ($n = 1.1 \times 10^6$, $m = 3.0 \times 10^6$), consistent with the setup in [25], where each vertex represents a user and each edge denotes a friendship. We also ensure that each $k$-clique has at least 10% coverage (i.e., $\tau = 1$), consistent with the method in [25]. We compare KCC with two existing methods: the Maximal Clique Cover (MCC) and the Enhanced Maximal Clique Cover (EMCC) [25]. Note that since we do not have access to the source code of MCC and EMCC, we directly use their reported results for comparison. While both MCC and EMCC effectively reduce advertisement costs by covering cliques with fewer vertices, our approach is superior. As shown in Table 2, KCC achieves the desired 10% coverage with fewer vertices than MCC and EMCC when $k \geq 4$. Specifically, KCC reduces the target set size by an additional 20% compared to MCC. Furthermore, KCC has a significantly faster runtime, more than 30 times faster than MCC and EMCC. Note that the percentage refers to the ratio of the cover size to the total number of vertices in the graph. Overall, our $k$-clique cover-based group buying advertising approach demonstrates superior performance in terms of both effectiveness and efficiency. Furthermore, our approach allows for customized clique sizes, making it particularly useful for managing group deals that require a specific number of participants.

## 7 Conclusion

In this paper, we formulate and study the $k$-clique $\tau$-cover problem. We also prove the NP-hardness of finding the minimum $k$-clique $\tau$-cover. We present an improved approach to compute a small cover based on the $k$-clique listing. In addition, we propose an efficient hierarchical solution to compute a small cover without listing the $k$-cliques. The experiments on ten real-world graphs verify the efficiency of our hierarchical solution. Several potential research directions remain open, such as extending our hierarchical solution to other subgraph models.

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

# A Dynamic Update

## A.1 Method

In this section, we explore how to dynamically update the $k$-clique $\tau$-cover when the graph changes. A naive method is to recompute the $k$-clique $\tau$-cover for the updated graph using the KCCB algorithm. However, this method does not utilize the existing cover.

Our approach involves materializing and maintaining a sequence of vertex sets $\mathcal{R}$ for the $k$-clique cover hierarchy. When the graph changes, we first update the original graph and determine the influenced area. For inserted edges, we update the vertex cover to cover these new edges. For deleted edges, we examine the adjacent vertices in the vertex cover and remove any redundant vertices from it. Next, we pass the updated vertex cover to the next layer and construct the induced subgraph. The difference between this induced subgraph and the one constructed by the materialized vertex set will be treated as the graph update. This process is repeated until we reach the last layer. The time complexity is bounded by $O(k \cdot m)$ and the space complexity is bound by $O(k \cdot n)$.

## A.2 Experimental Evaluation

We compare the efficiency of dynamically updating the $k$-clique cover hierarchy with recomputing the hierarchy. We evaluate the algorithms for the representative case where $\tau = 1$, by randomly removing 10% of the edges from the graph, then adding them back

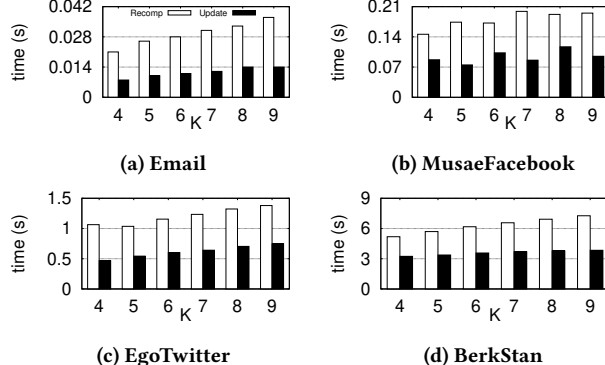

(a) Email          (b) MusaeFacebook

(c) EgoTwitter          (d) BerkStan

**Figure 11: Cumulative update time.**

and recording the cumulative update time. Figure 11 reports the cumulative update time of the algorithms. We can see that the dynamic update is about two times faster than the recomputation. For example, in the Email dataset, the dynamic update takes 0.01 seconds on average, whereas the recomputation takes around 0.03 seconds.

