# OpenReview forum: "Covering K-Cliques in Billion-Scale Graphs"
_ACM.org/TheWebConf/2025/Conference — WWW 2025 Poster_

### Official Review · Reviewer_dtdd · 2024-11-28

**Novelty:** 5
**Technical Quality:** 5

**Review:**

This paper introduces the $k$-clique $\tau$-cover problem, a generalization of the vertex cover problem, to efficiently cover $k$-cliques in large-scale graphs. The authors propose a scalable hierarchical algorithm with $O(k \cdot m)$ complexity, avoiding explicit $k$-clique enumeration. Pruning techniques, such as core number and graph coloring, further improve efficiency. Experiments show significant reductions in runtime and cover size compared to existing methods, demonstrating its potential for real-world applications like group-buying advertising.

Strengths:

1. The $k$-clique $\tau$-cover problem extends the vertex cover problem, providing a novel perspective on high-order graph structures in large-scale graph mining.

2. The proposed hierarchical algorithm is highly scalable, avoiding $k$-clique enumeration and incorporating effective pruning strategies like core number and graph coloring.

3. Experiments on real-world datasets demonstrate significant improvements in runtime and cover size compared to existing methods, highlighting the algorithm's practical relevance and scalability.

Weaknesses:

1. While the problem definition is novel, the hierarchical approach and pruning strategies rely on well-known techniques like core number and graph coloring, offering limited methodological innovation.

2. The paper lacks a thorough theoretical analysis of the algorithm's approximation guarantees or performance bounds, which are crucial for evaluating solution quality.

**Questions:**

The proposed hierarchical algorithm demonstrates strong empirical performance on real-world datasets, as shown in the experimental results. However, the paper lacks a theoretical analysis of the algorithm's approximation guarantees, optimality, or performance bounds. Could you provide more insights into the theoretical aspects of your method? Additionally, how do you anticipate the algorithm would perform on graphs with different structures (e.g., sparse versus dense graphs)? Would it be possible to adapt the algorithm or its parameters to optimize performance for different types of graphs?

**Reviewer Confidence:**

2: The reviewer is willing to defend the evaluation, but it is likely that the reviewer did not understand parts of the paper

**Scope:**

4: The work is relevant to the Web and to the track, and is of broad interest to the community

---

### Official Review · Reviewer_qEwj · 2024-12-02

**Novelty:** 4
**Technical Quality:** 4

**Review:**

### Summary
This paper addresses the problem of k-clique covering in large-scale graphs. The authors prove that the problem is NP-hard, and propose several approximation algorithms for the problem. They conducted experiments to check the effectiveness of the proposed methods in terms of efficiency.


### Strong points
* S1. The authors explore the k-clique t-cover problem.
* S2. They propose an efficient hierarchical method for the target problem.
* S3. They conduct extensive experiments on diverse real-world graphs in terms of efficiency.

### Weak points
* W1. The paper is not easy to follow and needs to enhance the presentation quality.
* W2. The scalability analysis is missing in the experiment.
* W3. The proposed method is based on approximation, but there is insufficient analysis regarding the correctness of the resulting output.

**Questions:**

* Q1. The explanation of the core ideas regarding the technical challenges addressed by each proposed method and how they were solved is unclear.
* Q2. It is difficult to easily grasp the importance of the problem addressed in the paper. It would be helpful to emphasize why efficiently solving this problem in large-scale graphs is important, beyond just explaining it in text, by highlighting its significance with some illustrations.
* Q3. The paper proposes KCCB and KTCC, but it is unclear when and which one should be used. What are the specific scenarios or conditions that determine the appropriate choice between them?
* Q4. How does the time and memory usage of the proposed method change with variations in the number of vertices and edges?

**Reviewer Confidence:**

2: The reviewer is willing to defend the evaluation, but it is likely that the reviewer did not understand parts of the paper

**Scope:**

3: The work is somewhat relevant to the Web and to the track, and is of narrow interest to a sub-community

---

### Official Review · Reviewer_fLvT · 2024-12-02

**Novelty:** 5
**Technical Quality:** 4

**Review:**

The paper explores the $k$-clique $\tau$-cover problem in graphs and introduces a hierarchical solution that calculates a compact cover without needing to enumerate  k -cliques. It makes a solid contribution by clearly defining the  k -clique \tau-cover problem and proving its NP-hardness.

pros:
1) The paper proposes an innovative hierarchical algorithm that avoids the computationally expensive process of  $k$-clique enumeration.
2) It is built on a strong theoretical foundation, including a proof of NP-hardness and efficient pruning strategies to enhance performance.

cons:
1) While the experiments use multiple public datasets, they mainly compare the two methods proposed by the authors, with little benchmarking against state-of-the-art algorithms. Additionally, the analysis of experimental results is not as comprehensive as it could be.
2) The handling of edge cases (e.g., sparse graphs or low  $k$-clique overlap) and the memory constraints of billion-scale applications are not sufficiently addressed. Limited evaluation of edge cases (e.g., sparse or highly overlapping cliques).

**Questions:**

1) In Algorithms 4 and 5, several heuristic methods are mentioned for obtaining the vertex cover. Do these different heuristics impact the algorithm’s time complexity?
2) In Section 6.1 on $k$-Clique Cover, the experiments only compare the time efficiency of KCCB and KCC. It would be helpful to include results from other advanced methods, such as the approach outlined in Reference 24.
3) In Experiment 6.1.2 on Cover Size, how exactly is the cover size calculated? The results show that KCC slightly outperforms KCCB on most datasets, which seems counterintuitive. Given that KCC involves pruning, one might expect its performance to be slightly worse. Could you provide a deeper analysis of why this occurs?
4) How does the hierarchical approach perform on extremely sparse graphs or graphs with a high density of overlapping cliques? Are there scenarios where the pruning strategies fail to deliver meaningful performance improvements?
5) In the case study, the proposed method is compared with Maximal Clique Cover (MCC) and Enhanced Maximal Clique Cover (EMCC) to demonstrate its effectiveness. However, the comparison only looks at the impact of different $k$ values on performance. Could you include experiments exploring how varying $\tau$-cover values affect results? Additionally, the case study focuses on group-buying advertising and uses the YouTube social network dataset. Why was this dataset chosen over others, such as the Amazon product co-purchasing network? Beyond group-buying advertising, what other domains could benefit from your approach? Could this method be extended or adapted to areas like biological networks, web graphs, or recommendation systems?

**Reviewer Confidence:**

3: The reviewer is confident but not certain that the evaluation is correct

**Scope:**

4: The work is relevant to the Web and to the track, and is of broad interest to the community

---

### Official Review · Reviewer_8yt7 · 2024-12-02

**Novelty:** 4
**Technical Quality:** 4

**Review:**

Summary：

This paper studies the 𝑘-clique 𝜏-cover problem, which is proved NP-hard. A basic solution established KCCB atop KClist is proposed. To improve the efficiency, a KCC algorithm is then presentd to avoid enumerating k-cliques. Finally, a KTCC algorithm is provided to extend KCC to 𝑘-clique 𝜏-cover problem. Some experiments are conducted to demonstrate the performance in terms of both runtime and cover size.

Strong points:

S1. The paper well-organized.

S2. Some experimental studies prove the efficiency of the proposed algorithms.

Weak points:

W1. The motivation is not clear.

W2. The experimental setting should be explained with more details.

W3. More discussions on effectiveness are required.

Detail comments:

D1. What real-world applications can the k-clique τ-cover problem be applied to, or what practical problems in real life can be modeled as a k-clique τ\tau-cover problem?

D2. The relationship among KCCB, KCC, KTCC should be clarified at least at the beginning of Sec-6. If I am correct, KCCB is utilized to solve both the k-clique cover problem with  τ=1 and k-clique  τ-cover prlblem with  τ>=2, while KCC is for k-clique cover problem and KTCC is for k-clique  τ-cover problem.

D3. Since k-clique  τ-cover is NP-hard, all algorithms tend to be approximate. Is it possible to evaluate the effectiveness w.r.t. exact k-clique cover or k-clique  τ-cover? If not, is it possible to provide some technical guarantee on the effectiveness, e.g., approximate ratio.

D4. From the aspect of cover size, it's hard to say KCC and KCCB have similar cover sizes except BerkStan dataset. So, it's better to clarify why KCC has a larger cover size than KCCB? Besides, the results of KCCB on billion-scale graphs are missing, the reasons should be explained. Is it because the issue of OOM or cannot return results within an acceptable time?

D5. The case study is not convince to me. Directly using the results in original papers is not a good choice. It's better to implement MCC and EMCC following the original papers.

**Questions:**

Please refer to the aforementioned weak points and detail comments.

**Reviewer Confidence:**

3: The reviewer is confident but not certain that the evaluation is correct

**Scope:**

3: The work is somewhat relevant to the Web and to the track, and is of narrow interest to a sub-community

---

### Official Review · Reviewer_QUsM · 2024-12-02

**Novelty:** 5
**Technical Quality:** 5

**Review:**

The paper discusses the problem of finding a set of nodes in the graph that cover all the cliques of a given size. The natural extension is to study if the set covers all the cliques of the graph by a factor of \tau for a given \tau. For interest, the computed set should be as small as possible. Trivially, a vertex cover is a set that covers all 2-cliques. The paper then proposes a hierarchical algorithm that samples from a cover for k-cliques to a cover for k+1-cliques. This works for \tau = 1. For a larger \tau, the paper extends the solution for the above case. The idea of using pruning based on degree/colors is common to clique enumeration algorithms.

The paper also shows the results of experiments on a collection of 10 graphs. Some minor comments:

1. The legend inside the plots is very difficult to read. Suggest the authors improve the readability of the legend.
Also, if the legend is common to all plots in a figure, then the legend may be moved to the top of the figure. Otherwise, the reader is forced to assume that the legend in plot (a) applies to all the plots in the figure.

2. The paper discusses a dynamic version of the problem. However, this has been pushed to the appendix completely. This is not a good idea and suggests that the authors fit most of the discussion on the dynamic case inside the paper. Since this section is not well developed, several questions are not addressed. See the section on Questions.

3.

**Questions:**

There are small aspects of the writeup that can be improved.

1. What do the names LL, LR, and SLL indicate. Do they refer to some intuitive mnemonics?

2. Provide more details of the dynamic approach. It may be good to provide results with the dynamic approach on all the graphs listed in Table 1.
3. In the dynamic algorithm, the sentence on "determine the influenced area" is not explained in detail.

**Reviewer Confidence:**

3: The reviewer is confident but not certain that the evaluation is correct

**Scope:**

4: The work is relevant to the Web and to the track, and is of broad interest to the community